# Inhibition of the Lysophosphatidylinositol Transporter ABCC1 Reduces Prostate Cancer Cell Growth and Sensitizes to Chemotherapy

**DOI:** 10.3390/cancers12082022

**Published:** 2020-07-23

**Authors:** Aikaterini Emmanouilidi, Ilaria Casari, Begum Gokcen Akkaya, Tania Maffucci, Luc Furic, Federica Guffanti, Massimo Broggini, Xi Chen, Yulia Y. Maxuitenko, Adam B. Keeton, Gary A. Piazza, Kenneth J. Linton, Marco Falasca

**Affiliations:** 1Metabolic Signalling Group, School of Pharmacy and Biomedical Sciences, Curtin Health Innovation Research Institute, Curtin University, Perth, WA 6102, Australia; cathemman@hotmail.com (A.E.); ilaria.casari@curtin.edu.au (I.C.); 2Centre for Cell Biology and Cutaneous Research, Blizard Institute, Barts and The London School of Medicine and Dentistry, Queen Mary University of London, London E1 2AT, UK; begum.akkaya@yahoo.com (B.G.A.); t.maffucci@qmul.ac.uk (T.M.); k.j.linton@qmul.ac.uk (K.J.L.); 3Prostate Cancer Translational Research Laboratory, Peter MacCallum Cancer Centre, Melbourne, VI 3000, Australia; luc.furic@monash.edu; 4Cancer Program, Biomedicine Discovery Institute and Department of Anatomy and Developmental Biology, Monash University, Clayton, VI 3800, Australia; 5Sir Peter MacCallum Department of Oncology, University of Melbourne, Parkville, VI 3010, Australia; 6Laboratory of Molecular Pharmacology, Istituto di Ricerche Farmacologiche Mario Negri IRCCS, 20156 Milan, Italy; federica.guffanti@marionegri.it (F.G.); massimo.broggini@marionegri.it (M.B.); 7Drug Discovery Research Center, USA Health Mitchell Cancer Institute, Mobile, AL 36604-1405, USA; xichen@health.southalabama.edu (X.C.); ymaxuitenko@health.southalabama.edu (Y.Y.M.); akeeton@health.southalabama.edu (A.B.K.); gpiazza@health.southalabama.edu (G.A.P.)

**Keywords:** prostate cancer, ABC transporter, ABCC1/MRP1, lysophosphatidylinositol, Docetaxel

## Abstract

Expression of ATP-binding cassette (ABC) transporters has long been implicated in cancer chemotherapy resistance. Increased expression of the ABCC subfamily transporters has been reported in prostate cancer, especially in androgen-resistant cases. ABCC transporters are known to efflux drugs but, recently, we have demonstrated that they can also have a more direct role in cancer progression. The pharmacological potential of targeting ABCC1, however, remained to be assessed. In this study, we investigated whether the blockade of ABCC1 affects prostate cancer cell proliferation using both in vitro and in vivo models. Our data demonstrate that pharmacological inhibition of ABCC1 reduced prostate cancer cell growth in vitro and potentiated the effects of Docetaxel in vitro and in mouse models of prostate cancer in vivo. Collectively, these data identify ABCC1 as a novel and promising target in prostate cancer therapy.

## 1. Introduction

Prostate cancer is the most common solid cancer in men [1]. To date, androgen deprivation therapy remains the most widely used treatment for prostate cancer, including late-stage cases [2]. Most advanced prostate cancers, however, will progress into a castration-resistant (also called androgen-refractory) state and 90% of men with castration-resistant prostate cancer will develop bone metastasis [1,2]. Despite advances in treatment efficiency, reduction of mortality rates and increased patient survival, prostate cancer remains the most common non-cutaneous malignancy and the third cancer-related cause of death among men in developed countries [3]. In this respect, identification of molecules and proteins involved in prostate cancer development and progression is critical in order to find potential targets and to develop novel and more active drugs.

ATP-binding cassettes (ABC) transporters are a large and diverse family of primary-active transporters that are present in all organisms and are well conserved across mammals [4]. Many studies have reported elevated expression of ABC transporters at both mRNA and protein levels in a variety of human cancers, such as breast, prostate and lung cancer [5]. Moreover, it was found that ABC transporters were not only highly expressed in solid tumours compared to normal tissues, but they also correlated with tumour grading. ATP Binding Cassette transporter C1 (ABCC1, formerly MRP1) overexpression was reported to be associated with increased tumour size, differentiation and invasion in breast, liver and lung cancer [6,7]. ABCC1 was first identified as the mediator of multidrug resistance in the small cell lung cancer cell line H69AR [8]. ABCC1 is a multi-specific efflux transporter localised to the plasma membrane where it can confer resistance to chemotherapeutic drugs such as doxorubicin, etoposide, and vincristine in vitro by transporting them extracellularly and therefore preventing intracellular accumulation of toxic levels of drug [9]. Indeed, in vivo expression of ABCC1 correlates with drug resistance and/or poor prognosis in a range of cancers [10]. Importantly, it has been demonstrated that in normal tissue ABCC1 can efflux endogenous leukotriene and prostaglandin eicosanoids (derivatives of long chain fatty acids), conjugated or co-transported with glutathione, and also steroids such as sulphated or glucuronidated estradiols [6]. ABCC1 is also reported to transport another lipid derivative, sphingosine-1-phosphate (S1P) [11]. The ability of ABCC1 to transport these bioactive lipids and steroids might therefore suggest additional roles for the transporter during cancer development and progression, beyond chemotherapeutic drug resistance.

Our group previously demonstrated that ABCC1 is involved in an autocrine loop by which prostate cancer cells can stimulate their own proliferation through the release of the lysophospholipid lysophosphatidylinositol (LPI) and activation of the G protein-coupled receptor GPR55 [12]. Specifically, the observation that siRNA-mediated downregulation of ABCC1 significantly decreased LPI export from prostate cancer PC3 cells pointed to a central role for the transporter in this process. Interestingly, S1P, which can be exported by ABCC1, is able to mediate proliferation and migration of breast cancer cells [13,14].

The ability of specific members of the ABC transporter family to secrete bioactive lipids into the extracellular medium to drive tumour progression [15,16] makes them novel potential targets for the development of anti-cancer therapeutics. In this study, we investigated the effect of ABCC1 blockade on prostate cancer cell proliferation in vitro and in vivo. Our results demonstrate the therapeutic potential of the pharmacological inhibition of ABCC1.

## 2. Results

### 2.1. ABCC1 Can Efflux LPI

We previously reported that prostate cancer PC3 cells are able to release LPI, which in turn stimulates a signalling cascade promoting their proliferation [12]. The demonstration that siRNA-mediated downregulation of the ABC transporter ABCC1 prevented LPI release and inhibited LPI-dependent cell signalling and proliferation led us to hypothesise that an autocrine loop exists by which prostate cancer cells release LPI via ABCC1 to sustain their proliferation through activation of GPR55 [12]. Consistent with this, expression of both ABCC1 and GPR55 is detected in human prostate cancer cell lines PC3, DU145 and LNCaP (Figure 1).

To confirm the role of ABCC1 as an LPI transporter, we transiently expressed ABCC1 in HEK293T cells. This cell line was selected because it has been used by many groups as a naïve line in which to study these transporters following transient or stable expression. In particular, it has been reported that the endogenous levels of ABCC1 are not detectable in these cells [17]. Consistent with this, we observed that expression of the transporter was clearly detected by Western blot in whole cell lysates prepared from cells transfected with a plasmid encoding for ABCC1 but not the naïve HEK293T cells or cells transfected with the empty vector (Figure 2A). Membrane vesicles were prepared and incubated in a reaction buffer containing ATP and the ligand of interest in order to measure the ability of the transporter to transfer the ligand from the buffer into the lumen of inside-out vesicles (only in this fraction of vesicles will the normally cytosolic nucleotide-binding domains be exposed to the added ATP). The in vitro assay was validated by measuring the transport of radiolabelled estradiol 17-β-D-glucuronide (^3^H-E2-17βDG), a known transport ligand of ABCC1 [6]. Data confirmed increased accumulation of ^3^H-E2-17βDG in vesicles from ABCC1-expressing HEK293T cells compared to vesicles derived from untransfected cells, describing Michaelis–Menten kinetics with a Km of 266 nM ^3^H-E2-17βDG (Figure 2B). Accumulation was inhibited by the ATPase inhibitor vanadate (Figure 2C), confirming a primary-active mechanism of transport, as expected for an ABC transporter. Transport of ^3^H-E2-17βDG was also inhibited by the ABCC1-specific inhibitor MK571 (Figure 2C), further validating the expression of functional ABCC1 in the in vitro assay.

To test whether LPI is a transport substrate of ABCC1, ^3^H-LPI was purified by fractionation and thin-layer chromatography from culture medium of ^3^H-inositol-labelled PC3 cells and used in the in vitro transport assay. Our data indicated that membrane vesicles prepared from ABCC1-expressing cells accumulated significantly more ^3^H-LPI than vesicles prepared from untransfected cells (Figure 2D). Transport of ^3^H-LPI was inhibited by vanadate and MK571 (Figure 2D), confirming a specific role for ABCC1 in this process. To further characterise the mechanism of ABCC1-induced transport of LPI, 100 μM glutathione was added to the reaction mixture as glutathione is known to stimulate transport of some ABCC1 ligands. No increase in LPI transport was detected upon glutathione supplementation (Figure 2D).

Taken together these data demonstrate that ABCC1 can efflux LPI directly, requiring neither conjugation nor co-transport with glutathione. This confirms that ABCC1 has a key role in the LPI-dependent autocrine loop.

### 2.2. ABCC1 Inhibitors Reduce Prostate Cancer Cell Growth without Affecting Normal, Immortalised Prostate Cells

To determine whether pharmacological inhibition of ABCC1 can affect prostate cancer cell proliferation through blockade of the ABCC1-mediated LPI autocrine loop, a panel of prostate cancer cell lines (PC3, LNCaP and DU145), as well as an immortalised normal prostatic epithelial cell line (PNT2), were treated with the ABCC1 inhibitors Reversan (Figure 3A) and MK571 (Figure 3B) and cell numbers were assessed after 72 h. Dose-response curves indicated that all three cancer cell lines were more sensitive to Reversan compared to the normal PNT2 cell line, with all cancer cells showing a statistically significant reduction in cell numbers upon treatment with 10 μM Reversan (Figure 3A). Similarly, MK571 appeared to reduce numbers of PC3 and LNCaP specifically, although values only reached statistical significance for LNCaP cells (Figure 3B). Overall, data further indicated that Reversan reduced cell numbers more efficiently than MK571.

Importantly, addition of exogenous LPI was able to counteract the effect of Reversan, resulting in increased LNCaP (Figure 4A) and PC3 (Figure 4B) cell numbers compared to cells treated with the inhibitor alone. On the other hand, exogenous LPI did not affect cell numbers significantly in the absence of the inhibitor (Figure 4A,B).

These data indicate that pharmacological inhibition of ABCC1 inhibits prostate cancer cell growth in vitro by blocking the ABCC1-mediated LPI autocrine loop.

Furthermore, we observed that Reversan inhibited both anchorage-dependent (Figure 5A) and anchorage-independent (Figure 5B) growth of HiMYC cells, a prostate cancer cell line derived from a mouse model that is genetically engineered to exhibit prostatic intraepithelial neoplasia and invasive malignancy upon overexpression of c-MYC [18]. Importantly, a direct correlation between c-MYC overexpression and ABCC1 upregulation has been observed in breast cancer [19] and aberrant expression of c-MYC has been detected in a number of human malignancies including prostate epithelial neoplasia [20,21]. All prostate cancer cell lines used in this study express c-MYC (Appendix A), consistent with previous data [22,23]. These observations, together with our data from HiMYC cells, prompted us to investigate the potential role of c-MYC in the regulation of the LPI-dependent, ABCC1-mediated proliferative autocrine loop. To this end, PNT2 cells were transfected with a plasmid encoding for c-MYC or the corresponding empty plasmid (pcDNA). Efficient overexpression was confirmed by quantitative real time PCR (Appendix A). First, we observed that overexpression of c-MYC increased the numbers of PNT2 cells (Figure 5C). Consistent with data in Figure 3A, Reversan did not affect growth of PNT2 cells transfected with pcDNA (Figure 5D). On the other hand, Reversan strongly reduced the number of cells overexpressing c-MYC (Figure 5D), indicating that, in these cells, ABCC1 inhibition was able to reduce c-MYC-induced cell growth specifically. Moreover, addition of exogenous LPI counteracted the effect of Reversan in c-MYC-expressing cells without significantly affecting the number of pcDNA-transfected cells (Figure 5D).

Taken together these data suggest that the ABCC1/LPI pathway regulates cell growth induced by c-MYC overexpression in immortalised prostate cells.

### 2.3. Combination of Sub-Optimal Concentrations of Reversan and Docetaxel Inhibits Prostate Cancer Cell Growth In Vitro and In Vivo

We next observed that treatment with sub-optimal concentrations of Reversan potentiated the effect of low doses of Docetaxel in LNCaP (Figure 6A) and PC3 (Figure 6B) cells. In addition, the combination of the two drugs strongly reduced HiMYC cell numbers (Appendix A) and the ability of these cells to form colonies in a clonogenic assay (Figure 6C), suggesting that ABCC1 inhibition can sensitize prostate cancer cells to the effect of Docetaxel. Consistent with this, treatment with Docetaxel and Reversan reduced the number of HiMYC colonies more potently than each treatment alone, as assessed by soft agar assay (Figure 6D). Importantly, addition of exogenous LPI partially rescued the inhibition induced by combination of the cytotoxic drug with the ABCC1 inhibitor (Figure 6D).

We then decided to determine the efficacy of Reversan alone or in combination with Docetaxel on prostate cancer growth in vivo. Experiments were performed by inoculating PC3 cells either orthotopically or subcutaneously in male athymic nude-Foxn1^nu^ mice. Mice injected subcutaneously were treated with distinct concentrations of Docetaxel or vehicle (control) in order to determine the most appropriate concentration to be used in combination (Appendix A). Based on these preliminary experiments, a dose of 3 mg/kg was chosen as a sub-optimal concentration of Docetaxel to combine with Reversan. PC3 cells were then inoculated orthotopically and mice were randomised by body weight into four groups of eight, 19 days post-inoculation (Study Day 0). Animals in each group were treated with a combination of Vehicle Control 1 (10% NMP/90% PEG 300)/Vehicle Control 2 (0.85% ethanol in saline), Docetaxel monotherapy (3 mg/kg), Reversan monotherapy (20 mg/kg) or a combination of Docetaxel/Reversan (3/20 mg/kg). Vehicle Control 1 and Reversan were administered for five consecutive days followed by two days rest each week over a four-week period. Treatments were withheld from individual animals where body weight loss exceeded the ethical limit until body weight recovered to greater than 85% of initial weight. Due to declining body weight of mice treated with the combination of Reversan and Docetaxel, this study was interrupted. Although Reversan proved unsuitable for these combinational experiments in vivo, our in vitro investigation still supported the conclusion that an ABCC1 inhibitor would potentiate the effect of Docetaxel. We therefore sought to identify novel ABCC1 inhibitors that might be more tolerable and might be used in combination with Docetaxel in vivo.

### 2.4. Identification of Novel ABCC1 Inhibitors and Validation of Their Activity In Vitro

It has been reported previously that Sulindac sulfide, the main metabolite of the non-steroidal anti-inflammatory drug Sulindac, is able to inhibit ABCC1 [24]. In order to develop novel potential ABCC1 inhibitors deprived of any anti-inflammatory activities, we synthesised a series of Sulindac derivatives that lacked inhibitory activity against cyclooxygenase. First, we tested a series of such derivatives on PNT2 and LNCaP cells and we identified a lead compound, coded as S3 (MCI-715), as the most effective in reducing the numbers of LNCaP cells with lower activity on normal epithelial PNT2 cells (Appendix A). S3 contains an indene scaffold identical to Sulindac, reported previously to inhibit ABCC1 [24], and is chemically related to a previously reported analogue, ADT-094, which has different biological activities [25].

To determine whether S3 inhibited LNCaP growth by affecting the ABCC1-dependent pathway, we then investigated whether the compound was able to inhibit ABCC1 using a live cell, flow cytometric assay based on the transport of Calcein-AM [26]. Calcein-AM is a non-fluorescent, membrane permeant dye but once in the cytoplasm, the acetomethoxy (-AM) group is cleaved by intracellular esterases to release the green-fluorescent Calcein. ABCC1 is able to efflux Calcein-AM (and Calcein) from cells thus reducing the intracellular accumulation. HEK293T cells were co-transfected transiently with pcDNA-ABCC1 and pDsRed (in a 3:1 w/w ratio, respectively). pDsRed encodes red-fluorescent protein (RFP), which labels the transfected population (Appendix A). The cells were then incubated with Calcein-AM in the presence or absence of S3 or Sulindac. The cellular accumulation of Calcein was quantified by flow cytometry and the inhibitory activities of S3 and Sulindac were compared. Non-linear regression analysis showed that both S3 and Sulindac were effective inhibitors of ABCC1 achieving 85% ± 5% and 95% ± 10% inhibition respectively (*p* > 0.05). S3, however, was almost 10-fold more potent than Sulindac for the inhibition of ABCC1 (Figure 7A), displaying an IC_50_ of 12 µM (95% CI of 7–22 µM), compared to Sulindac IC_50_ of 106 µM (95% CI of 68–168 µM).

Once confirmed that S3 was able to inhibit ABCC1 activity, we performed a dose response analysis in additional prostate cancer cell lines and PNT2 cells. Consistent with data in LNCaP cells (Appendix A), S3 reduced the number of both HiMYC and PC3 cells (Figure 7B). Data also suggested that cancer cells were more sensitive to the inhibitor compared to PNT2 cells (Figure 7B). Treatment with increasing concentrations of S3 impaired the ability of DU145 (Appendix A), LNCaP (Appendix A) and PC3 (Appendix A) cells to form colonies in clonogenic assays. In addition, S3 inhibited HiMYC colony formation in soft agar (Figure 7C). Importantly, a combination of a low concentration of Docetaxel (1 nM) with S3 (2.5 µM) reduced the number of HiMYC cells more potently than each drug alone (Figure 7D).

To investigate the effect of the combination of S3 and Docetaxel in more detail, we assessed viability of prostate cancer cell lines upon treatment with two different concentrations of S3 and increasing concentrations of Docetaxel. CompuSyn analyses revealed that the combination of the two drugs had a synergistic inhibitory effect on viability of PC3 (Appendix A), LNCaP (Appendix A) and DU145 (Appendix A) cells. Taken together, these data indicate that S3 not only reduced prostate cancer cell numbers and colony formation in soft agar, likely through inhibition of ABCC1, but it also potentiated the effect of Docetaxel in vitro.

### 2.5. In Vivo Activity of ABCC1 Inhibitor S3

To investigate the efficacy of S3 in combination with Docetaxel in vivo, male athymic nude-Foxn1^nu^ mice were inoculated orthotopically with PC3 cells and treated with a combination of Docetaxel (3 mg/kg/day) and S3 (50 mg/kg/day) or each inhibitor alone. S3 used in combination with Docetaxel showed a clear trend towards increased inhibition compared to each single treatment, although data did not reach statistical significance (Figure 8A). As a high proportion of animals in the vehicle control group failed to develop detectable prostate tumours (3 out of 8), a subcutaneous model was chosen for subsequent studies (Figure 8B). Two weeks of oral daily treatment with S3 (50 mg/kg) did not affect tumour growth, confirming that S3 was not effective against prostate cancer growth in vivo when used as a single agent at this concentration. Docetaxel at 3 mg/kg/day induced a strong delay in the growth, with a maximal T/C × 100 (weights of tumours from Docetaxel-treated (T) mice/weights of tumours from vehicle-treated (C) × 100) of 30. The concomitant treatment of Docetaxel and S3 further reduced tumour growth, with a maximal T/C × 100 of 20. In addition, starting from day 28 from tumour implant, tumour weights of mice treated with the combination were statistically significantly lower than tumour weights of mice treated with Docetaxel only (Figure 8B). Such an increased activity was not due to increased toxicity, as treatment with S3 did not affect weights of mice compared to vehicle-treated mice (Appendix A). Although treatment with Docetaxel slightly decreased body weights, addition of S3 did not result in any further reduction (Appendix A).

In addition, we performed a pharmacokinetic study by measuring S3 levels in comparison with Sulindac in mouse plasma by LC-MS/MS. Our data indicate that after a single oral administration of 100 mg/kg, S3 reached plasma Cmax of 356 ± 93 µM, which is almost 30 times higher than the IC_50_ required for the inhibition of ABCC1 compared to its parental molecule Sulindac, reaching Cmax of 63 ± 15 µM, 0.6 times its IC_50_ (Appendix A).

Taken together these data indicate that treatment with S3 is able to potentiate the effect of Docetaxel in vivo.

## 3. Discussion

Despite the significant improvement of patients’ survival rates, prostate cancer remains one of the major leading causes of cancer-related death among men in the western world [1,2]. This is due to treatment failure due to drug resistance or recurrence after prostatectomy. ABC transporter overexpression in different cancers is recognized to correlate with treatment resistance and patient outcome [27,28,29]. Notably, elevated expression of ABCC1 was detected in prostate cancer samples compared to normal tissues [30]. Although the extrusion of drugs by ABC transporters in cancer and their involvement in drug resistance is a well-described phenomenon [31], the hypothesis of a direct involvement of the transporters in cancer progression through the efflux of bioactive lipid molecules such as LPI, which has many physiological and pathological functions [32,33,34,35], has been investigated less intensively [36,37]. Our previous study [12] demonstrated that LPI is synthesised by prostate cancer cells through the action of the enzyme phospholipase A2. Evidence from siRNA silencing suggested that LPI is then transported to the extracellular space by ABCC1 where, in turn, it activates its receptor GPR55. Several reports indicate that GPR55 plays a crucial role in cancer progression [12,38,39,40]. In the current study, transient expression of ABCC1 in a naive human cell line has confirmed that ABCC1 can export LPI across the plasma membrane. Our study further shows that targeting of ABCC1 with novel inhibitors reduces cancer cell growth in vitro and in vivo in preclinical models of prostate cancer.

We have shown previously that pancreatic cancer cells also efflux LPI which, in turn, activates GPR55 in a similar autocrine loop [35]. In pancreatic cancer, the closely related ABCC3 mediates LPI efflux [35,41]. ABCC3 expression has been found to be very low in prostate cancer specimens while ABCC1 is upregulated [30]. This demonstrates that ABCC1, but not ABCC3, plays a key role in prostate cancer progression. The fact that both ABCC1 and ABCC3 transport LPI and are inhibited by S3 is not surprising considering that they share 57% primary sequence identity (rising to 73% when amino acid similarities are included) and they are already known to have overlapping transport substrate specificity [42]. However, our previous observations in pancreatic cancer differ from the current study of prostate cancer models. Notably, S3 was particularly active as a single agent in vivo in different pancreatic cancer models [35], whereas in the present study, in spite of the anticancer activity shown in different in vitro models, we did not observe a decrease in tumour growth in our in vivo prostate model. On the other hand, we did find that S3 potentiates the activity of Docetaxel when used in combination in mouse models of prostate cancer. Since it has been shown that the resistance to Docetaxel is associated with ABCB1 but not ABCC1 [43,44], it seems unlikely that by inhibiting ABCC1 we are just increasing the cellular accumulation of Docetaxel as a consequence of the impairment of its drug efflux ability. It must be noted, however, that re-introduction of LPI opposed the effect of the S3/Docetaxel combination only partially, and therefore we cannot completely rule out the possibility that the combination of the two drugs exerts its effect via a combined inhibition of the ABCC1/LPI/GPR55 loop and of the ABCC1-mediated export of Docetaxel. A possible explanation for the increased activity of Docetaxel in conjunction with ABCC1 inhibition could be a decrease in LPI release and subsequent reduction of GPR55 activation. Further studies are now required to establish the exact mechanism beyond the potentiation of Docetaxel efficacy following combination with ABCC1 inhibition. It would be interesting to investigate whether the reduced LPI release or rather the potential accumulation of intracellular LPI upon ABCC1 inhibition impinge on the microtubule stabilizing or anti-mitotic properties of Docetaxel. It is also worth mentioning that ABCC1 inhibition would reduce GPR55 activation and we reported previously that inhibition of this receptor potentiates the efficacy of chemotherapy [40]. Intriguingly, data in the literature also suggest that intracellular localization rather than expression of ABCC1 might be crucial for its role in prostate cancer cells [45]. In particular, the transporter was found to be localized specifically in lipid rafts and prostasomes in PC3 and LNCaP but not in PNT2 cells. The well-known role played by lipid rafts and prostasomes in signal transduction underlines the functional implications of this observation [46,47] and might explain the different sensitivity to ABCC1 inhibitors that we detected in prostate cancer cells compared to PNT2 cells. On the other hand, different expression levels/localization of GPR55 might also affect the response to the inhibition of the ABCC1/LPI loop. Additional studies are clearly required to identify the precise mechanism accountable for the regulation of the ABCC1/LPI/GPR55 axis, ultimately responsible for the differential sensitivity to ABCC1 inhibition.

Another important aspect to consider is the identification of the specific factors regulating ABC transporters expression in cancer settings. Early evidence showed that p53 mutations are associated with increased ABCC1 expression in prostate cancer cells [48]. In addition, NOTCH1 signalling has been shown to increase ABCC1 expression in prostate cancer stem cells [49]. Interestingly, a compelling study revealed a positive correlation between ABCC1 and MYCN in neuroblastoma patients exhibiting *MYCN* amplification as well as in neuroblastoma cells engineered to overexpress the protein [28]. Furthermore, it has been demonstrated that ABCC1 is a direct c-MYC target in breast cancer and glioblastoma cells [19,50]. As MYC is commonly overexpressed in prostate cancer where it is recognized as an early driver of carcinogenesis [51], this raises the question of whether upregulation of ABCC1 might be regulated by c-MYC in prostate cancer cells. Our Western blotting analysis revealed different expression levels of c-MYC in PC3, LNCaP and DU145 cells consistent with previous studies [22,23]. On the other hand, these cells displayed similar levels of ABCC1 and were all sensitive to ABCC1 inhibition. These data might suggest that protein levels of c-MYC, although different between the three prostate cancer cell lines, are sufficient to promote ABCC1 expression and activation of the ABCC1/LPI pathway to a similar extent in these cells. Additional work, however, is required to definitely establish whether c-MYC regulates ABCC1 protein levels in prostate cancer cells. Importantly, we showed that c-MYC overexpression promoted growth of immortalised epithelial prostate cells in a mechanism involving ABCC1, as demonstrated by the observation that Reversan inhibited the c-MYC-driven but not the normal growth of these cells. Whether c-MYC overexpression modulates ABCC1 expression/intracellular localization or it affects LPI synthesis/release or it regulates GPR55 expression/activity in normal prostate cells remains to be clarified. Understanding the link between c-MYC and the ABCC1/LPI pathway would help clarifying whether c-MYC overexpression in prostate cancer is associated with a hyperactive ABCC1/LPI pathway and could potentially provide a criterion to identify patients who might benefit from combined Docetaxel/S3 treatment. However, without further corroboration of the potential link between c-MYC and the ABCC1/LPI pathway, any suggested mechanism of efficacy should remain speculative. Furthermore, our data suggest that S3 treatment provides the opportunity to use lower concentrations of Docetaxel with consequent reduction of unwanted side effects. S3 has already demonstrated a favorable toxicity profile [35], a noteworthy feature considering that previous attempts to target ABC transporters have failed because of off-target toxicity [31].

Taken together, our results indicate that ABC transporters have a specific and direct role in cancer by releasing bioactive signalling molecules, and therefore their inhibition could be beneficial not only to prevent or oppose drug resistance but also to reduce cancer progression, especially in combination with chemotherapeutics.

## 4. Materials and Methods

### 4.1. Plasmids

Wild-type ABCC1 cDNA encoded by a recombinant pcDNA3.1 plasmid (pcDNA3-ABCC1) was a kind gift from Prof. Susan PC Cole (Queen’s University, Ontario, Canada). pDsRed2-C1 (pDsRed) was from Clontech (Mountain View, California, CA, USA).

### 4.2. Transient Transfection of HEK293T Cells for Two Colour Flow Cytometry

HEK293T cells were cultured as adherent monolayers in Dulbecco’s modified Eagle medium (DMEM) high glucose (Thermo Fisher Scientific, Waltham, MA, USA) supplemented with 10% foetal calf serum (FCS). Briefly, 6.25 × 10^5^ cells were seeded onto a T25 tissue culture flask and double transfected 24 h later with a transfection mix prepared from 7.5 μg pcDNA3-ABCC1 and 2.5 μg pDsRed in a 20 μL volume of 2.5% glucose and 17 μg of linear 25 kDa PEI (Sigma-Aldrich; Gillingham, Dorset, UK), diluted in 5 mL growth medium. After a further 24 h, the culture was supplemented with butyric acid to a final concentration of 2 mM to stimulate transcription. The cells were harvested after a further 24 h.

### 4.3. Transport of Radiolabelled Ligands into Inside-Out Vesicles

Prostate cancer PC3 cells were fed [^3^H] *myo*-inositol to convert into ^3^H-LPI. The ^3^H-LPI released by the cells was extracted by acid medium extraction and isolated by thin layer chromatography [35]. Membrane vesicles were prepared from HEK293T cells using a method described previously [35]. Vesicles comprising 60 μg of total protein were incubated in a reaction mixture of 150 μL buffer containing 10 mM ATP (Sigma-Aldrich, Gillingham, Dorset, UK), 10 mM MgCl₂ (Sigma-Aldrich, Gillingham, Dorset, UK), 100 μg/mL creatine kinase (Roche, Burgess Hill, West Sussex, UK), 10 μM creatine phosphate (Roche, West Sussex, UK), 400 nM (or otherwise stated) estradiol 17-β-D-glucuronide (Sigma-Aldrich, Gillingham, Dorset, UK) with 40 nCi ^3^H-estradiol 17-β-D-glucuronide (^3^H-E2-17βDG; Perkin Elmer, Boston, MA, USA) or LPI (2 μM cold-LPI (Sigma-Aldrich, Gillingham, Dorset, UK) spiked with 0.5 nCi ^3^H-LPI) in the presence or absence of 100 μM vanadate, 10 μM MK571 or 100 μM glutathione (all from Sigma-Aldrich, Gillingham, Dorset, UK). The vesicles were incubated at 37 °C for 15 min. The reaction was stopped by adding 1 mL of ice-cold transport buffer (50 mM Tris-HCl, 250 mM sucrose, pH 7.5, from Sigma-Aldrich, Gillingham, Dorset, UK) and immediately filtered through cellulose nitrate filter discs (0.2 μm pore size, 25 mm diameter, Whatman; Fisher Scientific, Loughborough, Leicestershire, UK) using a 1225 Sampling Manifold (Millipore, Watford, Hertfordshire, UK) and washed four times with 3 mL of ice-cold transport buffer. The cellulose nitrate filter discs were recovered and placed into scintillation tubes with 5 mL of scintillation fluid Optiphase HiSafe 3 (Perkin Elmer, Boston, MA, USA). The radioactivity content of each sample was analysed in a Beckman LS 6000SC scintillation counter (Beckman Coulter, High Wycombe, Buckinghamshire, UK). Cellulose nitrate filter discs were found to bind a small but measurable amount of the ^3^H-LPI or ^3^H-E2-17βDG and this background radioactivity was subtracted from the measured radioactivity. Each experiment was performed in triplicate and at least three independent experiments were carried out. To normalize the data, ^3^H accumulation in ABCC1 vesicles was compared to vesicles prepared from untransfected cells and the data presented as a ratio: accumulation in ABCC^+ve^ vesicles divided by accumulation in ABCC^−ve^ vesicles. Statistical analysis was by one-way analysis of variance (ANOVA) followed by Tukey’s post hoc testing using GraphPad PRISM^®^ V5.0 software (GraphPad Software, San Diego, CA, USA).

### 4.4. Drug Transport Assay by Two-Colour Flow Cytometry

Transiently-transfected HEK293T cells (5 × 10⁶) were harvested in versene and incubated with Calcein-AM (0.1 μM; Thermo Fisher Scientific, Waltham, MA, USA) in 200 μL growth medium with 0 μM to 750 μM inhibitor (Sulindac or S3), for 20 min at 37 °C. Stock solutions of Sulindac and S3 were prepared in DMSO; the vehicle had no effect on the transport assay. The cells were then washed twice, pelleted (160× *g*) and resuspended in 0.5 mL ice-cold DMEM minus phenyl red and supplemented with 1% FCS. Analysis was performed using a FACScan flow cytometer (Becton Dickinson, Franklin Lakes, NJ, USA). The population was gated for 10,000 single cells of normal size and granularity. Calcein content was measured in the FL-1 (green) channel, and red fluorescence from the expressed DsRed was measured in the FL-2 channel. Flow cytometry data were acquired using CellQuest Pro Software version 5.2.1 (BD Biosciences, San Jose, CA, USA) and analysed using FlowJo (Tree Star; Ashland, OR, USA). ABCC1 transport activity was inferred from the fold difference in Calcein content of untransfected cells versus the transfected cells. To compare independent data sets, the fold difference was normalized to 100% activity in the absence of inhibitor. Statistical analysis of the dose response from three biological replicate experiments was by GraphPad PRISM^®^ V5.0 software (Graphpad Software, San Diego, CA, USA). To determine IC_50s_, log (inhibitor) was plotted against activity with curve fitting by non-linear regression analyses.

### 4.5. Cell Culture and Treatments

PNT2 (Cellbank Australia, Code 95012613), LNCaP (ATCC Cat# CRL-1740, RRID:CVCL_1379), PC3 (ATCC Cat# CRL-1435, RRID:CVCL_0035) and DU145 **(**ATCC Cat# HTB-81, RRID:CVCL_0105) cell lines were maintained in RPMI-1640 supplemented with 10% (v/v) heat-inactivated foetal bovine serum (FBS), 2 mM glutamine and 1% (v/v) of 1× penicillin/streptomycin (all reagents from Thermo Fisher Scientific, Waltham, MA, USA). Cells were cultured in a humidified environment at 37 °C with a 5% CO_2_ atmosphere. Cell media was changed every other day. HiMYC cells were cultured in RPMI supplemented with 10% (v/v) FBS, 2 mM glutamine, 1% (v/v) of 1× penicillin/streptomycin, and 1 nM testosterone. Isolation of these cells (also referred to as HM-5 cells) was described previously [23]. Where indicated, cells were treated with Reversan, MK751, Sulindac and Sulindac derivatives. Sulindac derivatives were synthesized and chemically related to ADT-094 as previously reported [36].

### 4.6. Cell Growth

#### 4.6.1. Cell Counting

Cells were treated with the appropriate inhibitors in complete media. After 72 h, cells were trypsinised and manually counted with a hemocytometer, while using Trypan blue to exclude dead cells. For rescue experiments, cells were incubated with LPI (10 µM, resuspended in methanol:chloroform 1:1), Reversan (10 µM, resuspended in DMSO) or a combination of the two. Control cells were treated with each respective vehicle (or the two vehicles for a combination of the two).

#### 4.6.2. MTT Assays

Cells were seeded in 96 wells (5000 cells/well). After 24 h, cells were treated with S3 and Docetaxel, as specified in Appendix A, for a further 72 h. Cell viability was assessed by determining the metabolic activity of live cells through incubation with MTT (Thiazolyl Blue Tetrazolium Bromide) for the last 3 h, as previously described [52]. Synergism analysis was performed using CompuSyn Version 1.0 software (ComboSyn, Inc., Paramus, NJ, USA) based on Chou–Talalay’s combination index (CI) method [53].

#### 4.6.3. Clonogenic Assay

Prostate cancer cell lines were plated in 6 well plates (500 cells/well) and incubated for 10–12 days in complete medium supplemented with the indicated concentrations of Docetaxel, Reversan or a combination of the two drugs. Control cells were treated with vehicle alone, DMSO. Colonies were then fixed with 4% paraformaldehyde and stained with crystal violet (0.01% in PBS). Colonies were visualized with ChemiDoc XRS+ System (Bio-Rad Laboratories, Hercules, CA, USA) and quantified with ImageJ software. Each experiment was performed in three biologically independent replicates.

#### 4.6.4. Anchorage Independent Growth

HiMYC cells (10,000 cells/well) were plated on soft agar and anchorage independent growth was assessed as previously described [32]. Colonies were visualized with ChemiDoc XRS+ System (Bio-Rad) and quantified with ImageJ software. Each experiment was performed in three biologically independent replicates.

### 4.7. Quantitative Real-Time PCR Analysis

Total RNA was extracted from cells using the GeneJET^TM^ RNA purification Kit (Thermo Fisher Scientific, Waltham, MA, USA) according to the manufacturer’s instructions. RNA concentrations were quantified using a Nanodrop-1000 (ND-1000) before the reverse transcription using Maxima Reverse Transcriptase (Thermo Fisher Scientific). Quantitative real-time PCR analysis was performed using the 2× Maxima SYBR green/fluorescein (Thermo Fisher Scientific) qPCR mix, carried out in triplicates using the 7500 Real-Time PCR System (Thermo Fisher Scientific). Analysis was done using the Applied Biosystems^®^ 7500 Real-Time PCR Detection and Software v1.4 with the mRNA levels normalized to GAPDH as a housekeeping gene.

### 4.8. Western Blotting

Cells were lysed in radioimmunoprecipitation assay (RIPA) buffer supplemented with Protease/Phosphatase Inhibitor Cocktail (Cell Signaling Technology, Danvers, MA, USA, #5872) and sonicated. Proteins were separated by SDS-PAGE and detected by Western blotting according to standard procedures using the following antibodies in BSA 3%/TBST: ⍺/β tubulin (Cell Signaling Technology, #2148), GPR55 (Cayman Chemicals, Ann Arbor, MI, USA, #10224), c-MYC (ABCAM, Cambridge, UK, #28842), ABCC1 (Figure 1: Cell Signaling Technology, #72202; Figure 2: Enzo Life Sciences, Exeter, UK ALX-801-007-C125).

### 4.9. In Vivo Studies

#### 4.9.1. Docetaxel Dose Response Experiments and Orthotopic Model

PC3 cells were inoculated either subcutaneously (dose response experiments) or orthotopically in male athymic nude-Foxn1^nu^ mice. For the orthotopic experiments (Vivopharm LLC-Hummelstown, PA, USA), mice were randomised, using a matched pair distribution method, by body weight into four groups of eight, 19 days post-inoculation (Study Day 0). PC3 inoculation was performed by harvesting PC3 cells by trypsinization, washing them twice in HBSS and then resuspending them in HBSS:Matrigel^TM^ (1:1 v/v). A cell count was performed and the final cell density was adjusted with HBSS:Matrigel^TM^ (1:1 v/v) to 1.5 x 10^8^ cells/mL. Animals were inoculated while under intraperitoneally injected anaesthesia (Ketamine (14 mg/mL)/Xylazine (1.2 mg/mL)). Prior to inoculation, the skin at the incision site was swabbed with topical povidone iodine solution and then ethanol, then an incision was made into the skin directly over the prostate. A needle was introduced into the prostate, where 5–6 μL of cell suspension, consisting of 7.5 × 10^5^ PC3 cells, was discharged. Mice were administered a 200 μL bolus dose of Buprenex (Buprenorphine HCl, 0.01 mg/mL) subcutaneously for pain relief at the time of surgery and the following day. Animals in each group were treated with a combination of Vehicle Control 1 (10% NMP/90% PEG 300)/Vehicle Control 2 (0.85% ethanol in saline), Docetaxel monotherapy (3 mg/kg), Reversan monotherapy (20 mg/kg) or a combination of Docetaxel/Reversan (3/20 mg/kg). Vehicle Control 1 and Reversan were administered for five consecutive days followed by two days rest each week over a four-week period. Treatments were withheld from individual animals where body weight loss exceeded the ethical limit until body weight recovered to greater than 85% of initial weight.

#### 4.9.2. Docetaxel and the S3 Combination in the Xenograft Model

Male athymic nude-Foxn1^nu^ mice (6 weeks old) were obtained from ENVIGO RMS Srl (Correzzana, Italy). They were maintained under specific pathogen free conditions, housed in isolated vented cages, and handled using aseptic procedures. The Istituto di Ricerche Farmacologiche Mario Negri IRCCS adheres to the principles set out in the following laws, regulations, and policies governing the care and use of laboratory animals: Italian Governing Law (D.lgs 26/2014; Authorisation n.19/2008-A issued 6 March 2008 by Ministry of Health); Mario Negri Institutional Regulations and Policies providing internal authorization for persons conducting animal experiments (Quality Management System Certificate—UNI EN ISO 9001:2015—Reg. N° 6121); the NIH Guide for the Care and Use of Laboratory Animals (2011 edition) and EU directives and guidelines (EEC Council Directive 2010/63/UE).

PC3 cells were maintained in RPMI-1640 supplemented with 10% FBS. Exponentially growing cells were detached, washed and resuspended in PBS for in vivo injections. For this purpose, 4 × 10^6^ cells/mouse in a volume of 200 µL were injected subcutaneously in immunodeficient mice. Tumour growth was measured with a Vernier calliper every two or three days, and tumour weights (mg = mm^3^) were calculated using the formula: (length (mm) × width (mm)^2^)/2.

When the tumours reached approximately 150 mm^3^, animals were treated with Docetaxel 3 mg/kg i.v. every 7 days for 3 doses (q7dx3), S3 50 mg/kg orally daily for two weeks, or with the combination of Docetaxel plus S3. The efficacy of the treatment was expressed as best tumour growth inhibition [T/C × 100 = (tumour weight mean of treated tumours/tumour weight mean of control tumours) × 100]. T/C (%) values were measured at the indicated days using the formula: weights of tumours from Docetaxel-treated (T) mice/weights of tumours from vehicle-treated (C) × 100.

Animal body weight was monitored biweekly together with a physical examination for the duration of the experiment. Statistical differences between the groups were assessed using Student’s *t*-test. Statistical significance was determined without correction for multiple comparisons, with alpha = 5.0%.

### 4.10. Pharmacokinetic Methods

Female C57BL/6 mice (for S3 PK) or male athymic nude mice (for Sulindac PK) were purchased from NCI Animal Production Program (Frederick, MD, USA). The animals were housed in a temperature-controlled room (20−24 °C) and maintained in a 12 h light/12 h dark cycle. Food and water were available *ad libitum*. Sulindac (10 mg/mL suspension in Maalox (CVS. Pharmacy, Inc., Woonsocket, RI, USA) and S3 (10 mg/mL solution in 0.5% CMC/0.25% Tween 80 in water) were administrated to mice by oral gavage at a dose of 100 mg/kg. Blood was collected from 3 mice at 1, 2, 4, or 8 h post treatment (Sulindac) or 0.5, 2 or 5 h post treatment (S3) into K_2_EDTA tubes, stored on ice, and subsequently centrifuged at 2400 rpm for 15 min at room temperature. Plasma was separated and kept below −20 °C until being assayed using the previously reported LC-MS/MS method [54]. Plasma concentrations were quantified using a calibration curve based on calibration standards prepared in drug-free plasma.

## 5. Conclusions

This study demonstrates that pharmacological inhibition of ABCC1 reduces prostate cancer cells growth in vitro and sensitizes them to Docetaxel treatment both in vitro and in vivo. These results support the conclusion that further preclinical studies are needed to investigate the effect of ABCC1 inhibitors in combination with chemotherapeutic agents that might result in development of new treatments for prostate cancer.

## Figures and Tables

**Figure 1 cancers-12-02022-f001:**
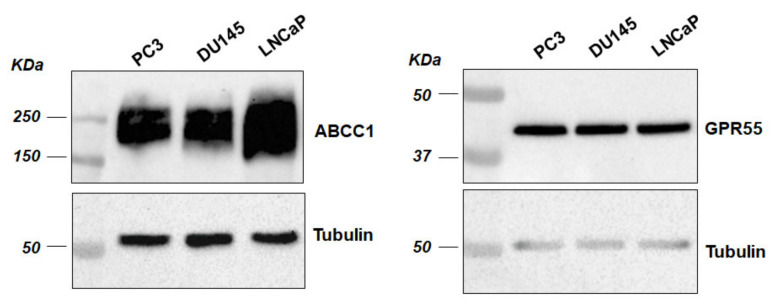
ATP Binding Cassette transporter C1 (ABCC1) and G protein-coupled receptor (GPR55) are expressed in prostate cancer cells. Expression of ABCC1 and GPR55 was assessed by Western blot analysis of lysates from the indicated prostate cancer cell lines. Membranes were stripped and re-incubated with anti-tubulin to confirm equal loading. The whole blot images can be found in Appendix A.

**Figure 2 cancers-12-02022-f002:**
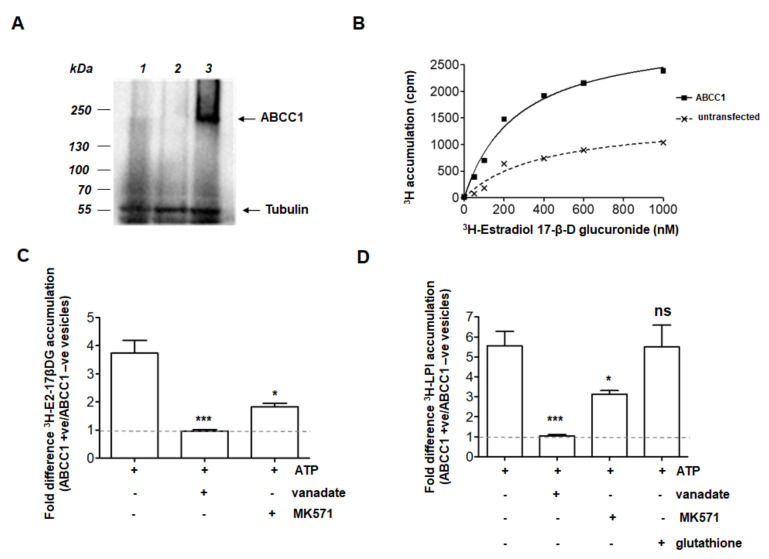
LPI is a transport substrate of ABCC1. (**A**) Expression of ABCC1 in HEK293T cells. Western blot probed with anti-ABCC1 antibody and anti-tubulin as loading control. Lane 1, whole cell lysate prepared from untransfected HEK293T; lane 2, whole cell lysate prepared from HEK293T cells transiently-transfected with pcDNA3.1; lane 3, whole cell lysate prepared from HEK293T transiently-transfected with pcDNA3.1-ABCC1. (**B**) ABCC1 transiently expressed in HEK293T cells is functional. Accumulation of ^3^H-estradiol-17-β-D-glucuronide (^3^H-E2-17βDG) in vesicles prepared from ABCC1-expressing cells (black squares, solid line) and untransfected cells (crosses, dashed line). Curves were fitted by non-linear regression analysis in GraphPad Prism. The curve fit describes Michaelis–Menten kinetics with a Km = 266 nM for ^3^H-E2-17βDG. (**C**) Transport of ^3^H-E2-17βDG by ABCC1 is sensitive to vanadate (100 μM) and MK571 (10 μM). The mean + SEM of the fold difference in accumulation of tritium between transporter-positive and transporter-negative vesicles is shown for 400 nM ^3^H-E2-17βDG. Means are compared to the transport efficiency in the absence of inhibitor by analysis of variance with Tukey’s post-test * *p* < 0.05 *** *p* < 0.001 (*n* ≥ 3). (**D**) ABCC1 transports LPI. Fold difference in ^3^H-LPI accumulation in the vesicles prepared from ABCC1-expressing cells compared to vesicles prepared from untransfected cells. The ^3^H-LPI transport activity of ABCC1 is inhibited by 100 μM vanadate or 10 μM MK571 but is not affected by 100 μM glutathione. Means were compared to the transport efficiency in the absence of inhibitor by analysis of variance with Tukey’s post hoc test *** *p* < 0.001, * *p* < 0.05, ns, not significant (*n* ≥ 3).

**Figure 3 cancers-12-02022-f003:**
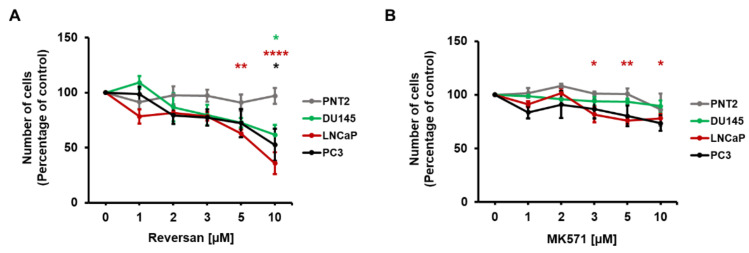
Pharmacological inhibition of ABCC1 reduces prostate cancer cell growth. Normal, immortalised epithelial prostate cells (PNT2) and prostate cancer cells (DU145, LNCaP, PC3) were treated with increasing concentrations of the ABCC1 inhibitors Reversan (**A**) and MK571 (**B**) or vehicle, dimethyl sulfoxide (DMSO) alone. Number of cells was assessed after 72 h by cell counting. Data are expressed as percentage of number of cells treated with DMSO (control) and are means ± SEM of *n* = 3 independent experiments performed in duplicate. For each cell line, one-way ANOVA with Dunnett’s multiple comparisons test was used for statistical analysis between each treatment and its corresponding control. Analysis was performed with GraphPad Prism version 6.0. * *p* < 0.05, ** *p* < 0.01, **** *p* < 0.0001.

**Figure 4 cancers-12-02022-f004:**
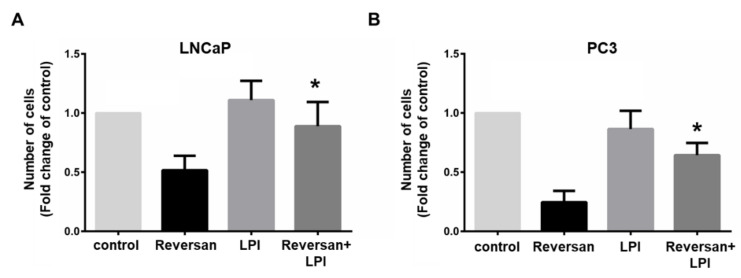
Exogenous LPI reverses the effect of Reversan on prostate cancer cells. Prostate cancer cell lines LNCaP (**A**) and PC3 (**B**) were treated with 10 μM Reversan, 10 μM LPI or a combination of the two. Control cells were treated with respective vehicles, as specified in the Materials and Methods section. Number of cells was assessed after 72 h by cell counting. Data are expressed as fold change of cells treated with each corresponding vehicle and are means + SEM of *n* = 3 independent experiments performed in duplicate. * *p* < 0.05 vs. Reversan.

**Figure 5 cancers-12-02022-f005:**
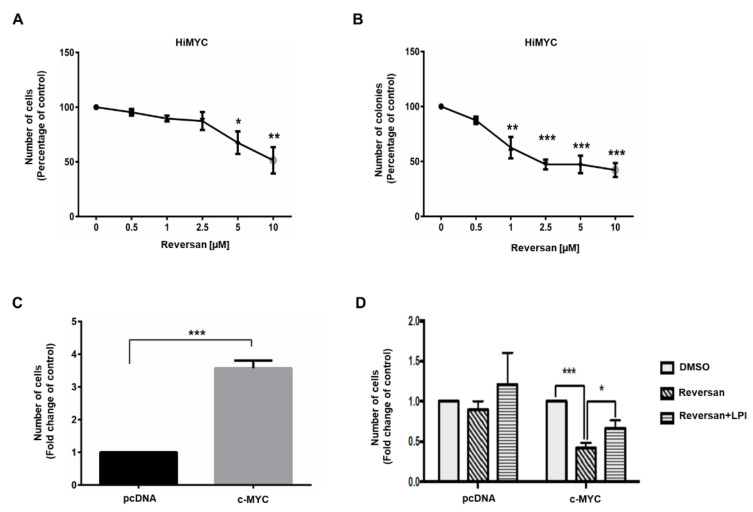
C-MYC regulates prostate cell growth in a mechanism involving ABCC1/LPI. (**A**) HiMYC cells were treated with the indicated concentrations of Reversan. Number of cells was assessed after 72 h by cell counting. Data are expressed as percentage of cells treated with vehicle, DMSO, and are means ± SEM from *n* ≥ 3 independent experiments. * *p* < 0.05, ** *p* < 0.01. (**B**) HiMYC cells were plated on soft agar and treated with increasing concentrations of Reversan. Number of colonies was assessed after 4 weeks. Data are expressed as percentage of cells treated with DMSO and are means ± SEM from *n* ≥ 3 independent experiments. ** *p* < 0.01, *** *p* < 0.001. (**C**) Immortalised epithelial prostate cells PNT2 were transfected with a plasmid encoding c-MYC or the corresponding empty vector (pcDNA). Number of cells was assessed after 72 h by cell counting. Data are expressed as fold change of pcDNA-transfected cells and are means + SEM from *n* = 3 independent experiments. *** *p* < 0.001. (**D**) PNT2 cells transfected with pcDNA or c-MYC-expressing plasmid were incubated with DMSO (control), Reversan (10 µM) or Reversan (10 µM) and LPI (10 µM) for 72 h. Data are expressed as fold change of each corresponding DMSO-treated cells and are means + SD of *n* = 4 independent experiments. * *p* < 0.05, *** *p* < 0.001.

**Figure 6 cancers-12-02022-f006:**
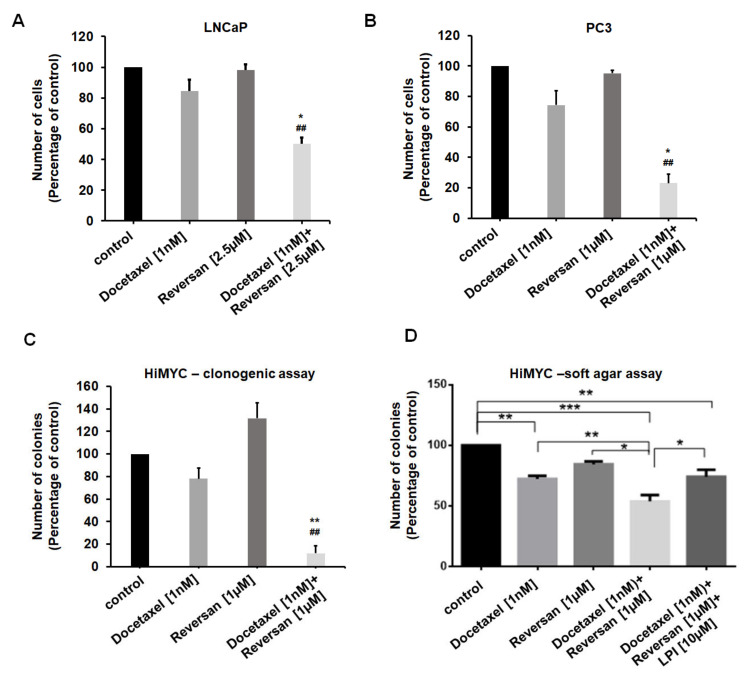
Effect of combinations of sub-optimal concentrations of Reversan and Docetaxel on prostate cancer cells growth. LNCaP (**A**) and PC3 (**B**) cells were treated with the indicated concentrations of Docetaxel or Reversan or combinations of the two drugs. Number of cells was assessed after 72 h by cell counting. * *p* < 0.05 vs. Docetaxel; ^##^
*p* < 0.01 vs. Reversan. (**C**) HiMYC cells were plated as single cells in a clonogenic assay and incubated in the presence of Docetaxel, Reversan or a combination of the two drugs at the indicated concentrations. Number of 2D colonies was assessed as specified in the Materials and Methods section. ** *p* < 0.01 vs. Docetaxel; ^##^
*p* < 0.01 vs. Reversan. (**D**) HiMYC cells were plated on soft agar and incubated in the presence of Docetaxel, Reversan or a combination of the two drugs at the indicated concentrations. Where specified, LPI was also added. Number of 3D colonies was assessed as specified in the Materials and Methods section. * *p* < 0.05, ** *p* < 0.01, *** *p* < 0.001. In all panels, data are expressed as percentage of results from cells treated with vehicle (control) and are means + SEM from *n* = 3 independent experiments.

**Figure 7 cancers-12-02022-f007:**
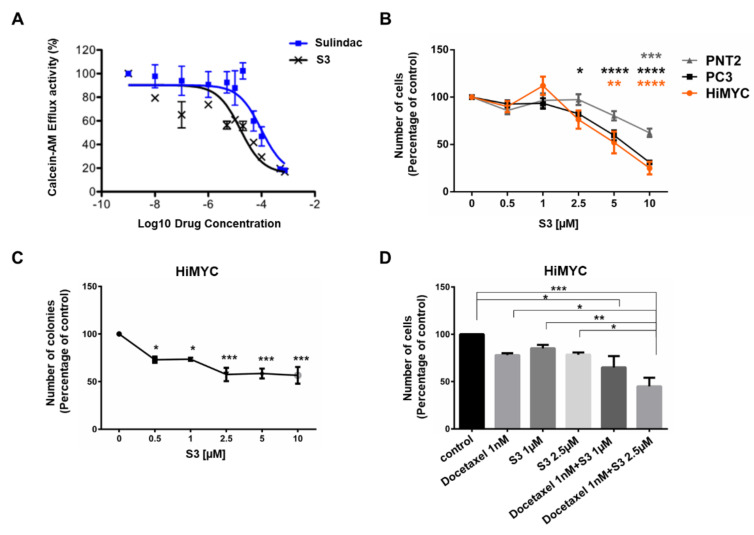
S3 inhibits ABCC1 function in vitro and prostate cancer cell growth and colony formation. (**A**) Results from in vitro assay measuring Calcein-AM efflux in the presence of increasing concentrations of S3 and Sulindac, indicating that S3 is 10-fold more potent than Sulindac for inhibition of the transport activity of ABCC1. (**B**) The indicated cell lines were incubated in the presence of increasing concentrations of S3 or vehicle, DMSO. Number of cells was assessed after 72 h by cell counting. Data are expressed as percentage of cells treated with vehicle (control) and are means ± SEM of *n* ≥ 3 independent experiments. * *p* < 0.05, ** *p* < 0.01, *** *p* < 0.001, **** *p* < 0.0001. (**C**) HiMYC cells were plated on soft agar and treated with increasing concentrations of S3. Number of colonies was assessed after 4 weeks using ImageJ software. Data are expressed as percentage of colonies from cells treated with DMSO (control) and are means ± SEM of *n* ≥ 3 independent experiments. * *p* < 0.05, *** *p* < 0.001. (**D**) HiMYC cells were treated with the indicated concentrations of Docetaxel and S3 alone or in combination. Cell counting with trypan blue exclusion was used to assess the number of cells after 72 h. Data are expressed as percentage of cells treated with DMSO (control) and are means + SEM of *n* ≥ 3 independent experiments. * *p* < 0.05, ** *p* < 0.01, *** *p* < 0.001.

**Figure 8 cancers-12-02022-f008:**
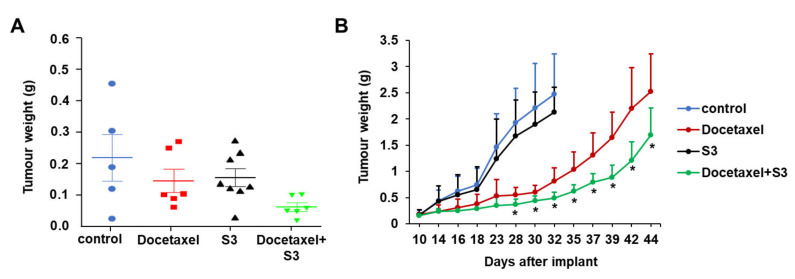
S3 potentiates the anti-tumour activity of Docetaxel in in vivo models of prostate cancer. (**A**) Male athymic nude-Foxn1^nu^ mice (*n* = 8) were inoculated orthotopically with PC3 cells and treated with a combination of Docetaxel (3 mg/kg/week) and S3 (50 mg/kg/day) or each inhibitor alone for 28 days. (**B**) PC3 cells were inoculated subcutaneously in nude mice. When tumours reached the designated size, mice were treated with Docetaxel (3 mg/kg), S3 (50 mg/kg) or a combination of the two drugs. Control mice were treated with vehicle. Tumour weights were measured at the indicated days. Data are means + SD from *n* = 7 mice/group. * *p* < 0.05.

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
