# Peer review of "Inhibition of the Lysophosphatidylinositol Transporter ABCC1 Reduces Prostate Cancer Cell Growth and Sensitizes to Chemotherapy"

_cancers, 2020, doi:10.3390/cancers12082022_

Round 1

Reviewer 1 Report

Emmanouilidi and co-authors presented a very nice work entitled: "Inhibition of the lysophosphatidylinositol transporter ABCC1 reduces prostate cancer cell growth and sensitizes to chemotherapy". I very much enjoyed reading the manuscript and the consistency maintained in terms of science and readability.

Comments:
Is there a mechanism of action that how at structural level (actually we know 'why') this ABCC1 transporter prevents entry doxorubicin, etoposide, and vincristine while it allows transport of bioactive lipids and steroids. I agree this is beyond chemotherapeutic drug resistance, but with curiosity is there something in the literature, to include in the introduction of the manuscript.

Lines 142 and 143: "Reversan reduced cell numbers more efficiently than MK571, with all three cancer cell lines showing a statistically significant reduction in cell numbers upon treatment with....." and Figure 2a and 2b. The activity of both compounds Reversan and MK571 for the cell line DU145, looks mostly similar?

Can author try to show the structure of synthesised Sulindac derivatives that were tested on PNT2 and LNCaP cells. Especially, the inhibitor S3 which is most promising one.

Author Response

Answers to Reviewer 1.

  1. Is there a mechanism of action that how at structural level (actually we know 'why') this ABCC1 transporter prevents entry doxorubicin, etoposide, and vincristine while it allows transport of bioactive lipids and steroids. I agree this is beyond chemotherapeutic drug resistance, but with curiosity is there something in the literature, to include in the introduction of the manuscript.

We have amended the text to clarify that accumulation of doxorubicin, etoposide, and vincristine in vitro actually depends on the same transport mechanism, i.e. intracellular accumulation is prevented because the drugs are transported extracellularly by ABCC1, similarly to LPI. Amended sentence in the text is as follow:”ABCC1 is a multi-specific efflux transporter localised to the plasma membrane where it can confer resistance to chemotherapeutic drugs such as doxorubicin, etoposide, and vincristine in vitro by transporting them extracellularly and therefore preventing intracellular accumulation of toxic levels of drug”.

  1. Lines 142 and 143: "Reversan reduced cell numbers more efficiently than MK571, with all three cancer cell lines showing a statistically significant reduction in cell numbers upon treatment with....." and Figure 2a and 2b. The activity of both compounds Reversan and MK571 for the cell line DU145, looks mostly similar?

We are very grateful for this comment as it allowed us to realise that there had been a mistake in the preparation of this Figure (the same curve for DU145 had been used inadvertently in the two graphs). We apologise for this mistake. The correct response of DU145 cells to MK571 treatment is now shown in the revised Figure 3B, confirming the reduced efficacy of this drug on all prostate cancer cell lines compared to Reversan. Please note that we also changed the y-axis, to express data as percentage of control rather than fold change. 

  1. Can author try to show the structure of synthesised Sulindac derivatives that were tested on PNT2 and LNCaP cells. Especially, the inhibitor S3 which is most promising one.

We certainly agree that it would be important to provide the structure of sulindac derivatives and S3 in the future, although premature disclosure in this manuscript would nullify any potential for patent protection of S3 that is essential for any possibility of clinical development. We have, however, added some information in the text, stating that S3 contains an indene scaffold identical to the NSAID, sulindac, reported previously to inhibit ABCC1 (Whitt et al, J. Biomed. Res. 30:120-133, 2016) and chemically related to a previously reported analogue, ADT-094, which has different biological activities (Li et al, Oncotarget, 6: 27403-27415, 2015). We added these citations to the revised manuscript and believe this is adequate information for the reader if there is the interest of developing analogues with improved potency, selectivity, and drug-like properties.

Reviewer 2 Report

The manuscript by Emmanouilidi et al. entitled "Inhibition of the lysophosphatidylinositol transporter 3 ABCC1 reduces prostate cancer cell growth and 4 sensitizes to chemotherapy" studied the role of ABCC1 and claimed that the  inhibition of that transporter reduces prostate cancer cell and sensitizes the chemotherapeutic agent docetaxel. The manuscript seems interesting however the in vivo combination study result seems very minor improvement of the activity of docetaxel when combined with ABCC1 inhibitor (Fig 7). In addition the manuscript lacks mechanistic data that support their claim. The authors need to perform additional experiment in in vivo tumor tissues to show how ABCC1 sensitizing docetaxel.

Major comments:

  1. The author should show the level of Docetaxel in the tumor tissues treated with docetaxel alone or in combination with S3 to determine whether S3 is inhibiting the transport of docetaxel out of the cells. Additionally it would be interesting to see the serum level of both drugs.
  2. Both in vitro  and in vivo data should be analyzed to provide that there is any synergistic effect with the combination to show the sensitization.  
  3. Please provide details of HiMyc cell isolation.
  4. Figure 4D showed that reversan does not have any effect before overexpression of cMyc. The author should provide the myc expression of all the studied prostate caner cell lines (LNCaP, DU145 and PC3) to show these cells have high level of Myc since they are sensitive to reversan and ABCC1 induced cell growth inhibition is mediated through Myc.

Author Response

Answers to Reviewer 2.

  1. The author should show the level of Docetaxel in the tumor tissues treated with docetaxel alone or in combination with S3 to determine whether S3 is inhibiting the transport of docetaxel out of the cells. Additionally it would be interesting to see the serum level of both drugs.

Although this is an interesting point, tumour tissues and serum from mice used in our in vivo experiment are not available anymore therefore this analysis would not be possible without repeating the experiment. Strict ethical requirements prevent us from simply repeating an experiment which has already been performed. In response to the Reviewer’s suggestion, we have measured plasma levels of S3 in mice following oral administration and added this pharmacokinetic data to the revised manuscript (Supplementary Figure 12). Interestingly, these new data show appreciable higher plasma levels of S3 compared with levels of sulindac sulfide generated from sulindac, which is consistent with greater antitumor efficacy of this class of compounds compared with sulindac as we have observed in other studies not reported here.

  1. Both in vitro and in vivo data should be analyzed to provide that there is any synergistic effect with the combination to show the sensitization.  

The effect of different combinations of S3 and Docetaxel has now been tested on three prostate cancer cell lines and combination indexes have been determined by CompuSyn. These data are presented in Supplementary Figures 8-10 and indicate that combination of the two drugs has a synergistic inhibitory effect on all three cell lines. Text has also been amended accordingly. As for the in vivo experiments, a proper estimation of synergistic effect would have required different doses for each single agent and for their combination. We have very restricted policy for animal experiments, which do not allow us to perform this kind of analysis. We believe, however, that the enhanced effect of S3 and Docetaxel combination in vivo is very clear, in particular in Figure 8B, where a sub-optimal concentration of S3 was used. The fact that treatment with such a low concentration of S3 alone had no effect on tumour growth clearly indicates that the effect of the two drugs together was not simply due to combined effects of each single drug.

  1. Please provide details of HiMyc cell isolation.

Isolation of these cell lines was reported previously. We have amended the Materials and Methods section to state this and we have added the corresponding reference.

  1. Figure 4D showed that Reversan does not have any effect before overexpression of cMyc. The author should provide the myc expression of all the studied prostate cancer cell lines (LNCaP, DU145 and PC3) to show these cells have high level of Myc since they are sensitive to reversan and ABCC1 induced cell growth inhibition is mediated through Myc.

We have added a blot showing the expression levels of c-MYC in the prostate cancer cell lines (presented in the new Supplementary Figure 1).

Reviewer 3 Report

Brief Summary: The aim of the study by Emmanouilidi et al., was to assess the therapeutic potential of ATP binding cassette transporter C1 (ABCC1) inhibition in prostate cancer (PCa) cells. The authors utilize an array of assays, amongst which are clonogenic, drug transport and in vivo xenograft models. The experiments were performed with the immortalized human prostatic epithelial PNT2 cells, the human PCa PC3, LNCaP and DU145 cells as well as, the murine-derived PCa cell line, HiMYC. Based on data from their previous publication, the authors confirm that ABCC1 acts to efflux lysophospholipid lysophosphatidylinositol (LPI) in PC3 cells. Moreover, they show that pharmacological ABCC1 inhibition reduces PCa cell growth with LPI reversing the phenotype. ABCC1 inhibition can also potentially enhance the therapeutic index of Docetaxel chemotherapy in the xenograft PC3 mouse model. In addition, they provide experimental data on the action of a novel ABCC1 inhibitor (S3) in PCa that suggests feasibility of therapeutically targeting ABC transporters in PCa. Several studies have examined the role of ABC transporters in drug resistance and conventional chemotherapy sensitization in the PCa setting. Although the data presented here can serve to promote further pre-clinical and co-clinical studies for the development of new PCa treatments with ABC transporter inhibitors, the authors should address some issues to elucidate their findings.

Strengths of the study:

  • Evidence for additional therapeutic activity of multidrug resistance transporter targeting in PCa.
  • Identification and initial characterization of a promising novel ABCC1 inhibitor for PCa treatment.
  • Use of several cell lines for majority of experiments.

Weaknesses of the study:

  • Lack of rigorous PCa cell line characterization, hindering the exact role of MYC in the ABCC1 inhibition response.
  • Use of only one PCa cell line for in vivo xenograft studies.

Specific Comments: The authors demonstrate that ABCC1 inhibition reduced PCa cell growth and facilitated chemotherapy sensitivity. However, there are several issues to be addressed in their study.

Abstract: Nice summary of the study outlining the objective, major findings and methods used as well as conclusion and significance.

Introduction: This section is well structured and balanced in terms of background information on PCa, ABC transporters with emphasis on the role of ABCC1 in cancer. The authors highlight the rationale of their study and the significance of their results.

Materials and Methods: The experimental procedures and study design are adequately explained, and details are properly provided.

Results and Figures: Overall results and figures are presented in orderly and comprehensible fashion. The authors describe the stud designs and data analysis. However, I have identified some issues that require further clarification. Comments:

  • In the section 2.1., ABCC1 is transiently overexpressed in the immortalized human embryonic kidney HEK293T cells to confirm its role as an LPI transporter. Since the ABC transporters are ubiquitously expressed in mammalian cells, how do the authors explain the lack of ABCC1 expression in the HEK293T cells, as seen in Figure 1A? [minor]
  • In Figure 1A, the loading control (e.g. beta-actin) and the indication of ABCC1 molecular weight are missing in the Western blot. [major]
  • The authors do not comment in the Results or the Discussion on the lack of glutathione-mediated LPI transport in the PC3 setting. The authors should provide some insights. What does this mean biologically for the mechanism of ABCC1 action? [minor]
  • In section 2.2., the authors test the dose-response of ABCC1 inhibition in several cell lines. However, they do not provide any data on the expression levels of ABCC1 in these cell lines, making it very difficult to correlate ABCC1 inhibition efficiency to actual ABCC1 expression. The manuscript would significantly benefit from a more detailed characterization of the cell lines used, at least in terms of target protein/gene expression. [major]
  • In Figure 4, the authors suddenly switch to the murine HiMYC cells attempting to investigate the potential role of Myc in ABCC1 inhibition responses. As mentioned above, the authors need to assess or at least discuss with relevance to published data, Myc expression in the human PCa cell lines they used for these and consequent experiments. Literature has shown that PC3 and LNCaP cells exhibit high Myc expression levels, whereas DU145 appear to have no/minimal Myc (Cassinelli et al., 2004, PMID: 15294455; Fan et al., 2016, PMID: 26279298; Itkonen et al., 2013, PMID: 23720054). The c-Myc overexpression experiments were performed only in the non-cancerous prostate cell line, PNT-2. The authors should assess Myc levels in their experiments and discuss why all the PCa cell lines respond in a similar manner to ABCC1 inhibition despite their difference in Myc expression. [major]
  • In Figure 4, RT-qPCR confirmation of c-Myc overexpression is missing. [minor]
  • In Section 2.3., the combination of Reversan (ABCC1 inhibitor) and Docetaxel has the most potent effect in reducing PCa cell growth. Have the authors tried to reverse that effect by adding LPI? This can further clarify the role of ABCC1 in chemotherapy sensitization [minor]
  • For the in vivo experiments, what was the rationale of choosing the PC3 cell line over the other human PCa cell lines? [minor]
  • In Section 2.4., the authors performed colony formation assays to assess the inhibitory activity of the novel S3 inhibitor, only in the murine HiMYC cell line. The study will benefit from similar experiments on the human PCa cell lines. [minor]

Discussion and Conclusion: The authors nicely discuss their findings and put them in context of published data. However, this section needs improvement. Comments:

  • The Conclusion section should come right after the Discussion and not after the Methods. [minor]
  • The authors comment on the lack of S3 therapeutic efficacy as a single agent their xenograft PC3 model, suggesting that it may be due to a decrease in LPI release rather than lack of Docetaxel intracellular accumulation. The rescue experiment suggested above in Section 2.3. may clarify this issue with experimental data. [minor]
  • The authors discuss the implication of Myc in the ABCC1 inhibition response. However, they can only speculate since they do not provide any direct evidence of Myc overexpression in their experimental setup using the human PCa cell lines. As mentioned above, a more detailed characterization of these cell lines in terms of ABCC1 and Myc expression would significantly improve the manuscript. [major]

Author Response

Answers to Reviewer 3.

  1. In the section 2.1., ABCC1 is transiently overexpressed in the immortalized human embryonic kidney HEK293T cells to confirm its role as an LPI transporter. Since the ABC transporters are ubiquitously expressed in mammalian cells, how do the authors explain the lack of ABCC1 expression in the HEK293T cells, as seen in Figure 1A? [minor]

ABC transporters are ubiquitously expressed in all species across both the animal and plant kingdoms but while there are 48 different ABC transporters in humans they are not ubiquitously expressed in all tissues. There are many cell lines that do not express the polyspecific ABC transporters ABCB1, ABCC1, ABCC3 or ABCG2 that might confuse the interpretation of our experimental results. We selected the HEK293(T) cell line because it has been demonstrated by many groups as a naïve line in which to study these transporters following transient or stable expression. For example, The Gottesman lab has demonstrated the absence of

ABCB1 and ABCG2, Robinson et al., Drug Metab Dispos 2019, 47(7): 715–723; The Iram lab has demonstrated the absence of ABCC1, Tan et al., Drug Metab Dispos 2018, 46 (12) 1856-1866; and the Kruh lab has demonstrated the absence of ABCC3, Zeng et al., Cancer Res 2000, 60(17):4779-84.

We have added information on ABCC1 (and corresponding reference) in the text

  1. In Figure 1A, the loading control (e.g. beta-actin) and the indication of ABCC1 molecular weight are missing in the Western blot. [major]

We have replaced the Western blot image in previous Figure 1A (Figure 2A in the revised manuscript). The new blot shows both ABCC1 and Tubulin (as loading control). Molecular weights have also been added as requested.

  1. The authors do not comment in the Results or the Discussion on the lack of glutathione-mediated LPI transport in the PC3 setting. The authors should provide some insights. What does this mean biologically for the mechanism of ABCC1 action? [minor]

As requested, we have amended the results section to comment that LPI itself is likely transported by ABCC1, unmodified by conjugation with glutathione. Amended sentence in the text reads: “Taken together these data demonstrate that ABCC1 can efflux LPI directly, requiring neither conjugation nor co-transport with glutathione. This confirms that ABCC1 has a key role in the LPI-dependent autocrine loop.”

  1. In section 2.2., the authors test the dose-response of ABCC1 inhibition in several cell lines. However, they do not provide any data on the expression levels of ABCC1 in these cell lines, making it very difficult to correlate ABCC1 inhibition efficiency to actual ABCC1. The manuscript would significantly benefit from a more detailed characterization of the cell lines used, at least in terms of target protein/gene expression. [major]

As requested, we have added a blot showing the expression levels of ABCC1 in PC3, DU145 and LNCaP (together with expression of GPR55, now presented in the new Figure 1). As the blot indicates, expression levels of ABCC1 is very similar between the three cell lines. However, we would like to point out that other factors, other than ABCC1 expression, could explain the effect of ABCC1 inhibition efficiency such as the expression of the LPI receptor GPR55 and ABCC1 localization. Indeed, a previous study (Goma’ et al. OncoTargets and Therapy 2014) reported that ABCC1 localization in PC3 and LNCaP is different from that observed in PNT2. In particular, a specific localization in lipid rafts and prostasomes has been observed in PC3 and LNCaP, but not PNT2, cells. The well known role played by lipid rafts and prostasomes in signal transduction underlines the functional implications of this observation.

  1. In Figure 4, the authors suddenly switch to the murine HiMYC cells attempting to investigate the potential role of Myc in ABCC1 inhibition responses. As mentioned above, the authors need to assess or at least discuss with relevance to published data, Myc expression in the human PCa cell lines they used for these and consequent experiments. Literature has shown that PC3 and LNCaP cells exhibit high Myc expression levels, whereas DU145 appear to have no/minimal Myc (Cassinelli et al., 2004, PMID: 15294455; Fan et al., 2016, PMID: 26279298; Itkonen et al., 2013, PMID: 23720054). The c-Myc overexpression experiments were performed only in the non-cancerous prostate cell line, PNT-2. The authors should assess Myc levels in their experiments and discuss why all the PCa cell lines respond in a similar manner to ABCC1 inhibition despite their difference in Myc expression. [major]

A shown in Supplementary Figure 1, we observed similar c-MYC levels in the three prostate cancer cell lines. This is consistent with other reports where, although reduced levels of c-MYC in DU145 are observed, the protein is clearly detectable both by Fan et al., 2016, (PMID: 26279298, Fig. 3B) and Rebello et al., 2016 (PMID: 27486174, Supplementary Figure 1A). LNCaP and PC3 but not DU145 are shown by Itkonen et al., 2013 (PMID: 23720054).

  1. In Figure 4, RT-qPCR confirmation of c-Myc overexpression is missing. [minor]

These data have been added and they are now presented in the new Supplementary Figure 2

  1. In Section 2.3., the combination of Reversan (ABCC1 inhibitor) and Docetaxel has the most potent effect in reducing PCa cell growth. Have the authors tried to reverse that effect by adding LPI? This can further clarify the role of ABCC1 in chemotherapy sensitization [minor]

We now present data showing the effect of adding exogenous LPI on cells treated with Docetaxel and Reversan. In particular, we determined the effect of combination of the drugs in the presence or absence of LPI on the number of HiMYC cells. These results (presented in the revised Figure 6D) show that LPI is able to rescue the inhibitory effect of Docetaxel and Reversan partially. Panel C of the original Figure 5 is now presented in Supplementary Figure 3

  1. For the in vivo experiments, what was the rationale of choosing the PC3 cell line over the other human PCa cell lines? [minor]

The PC3 cell line is extensively used in xenograft models, as it does not pose major challenges during implantation and tumour growth. We have used it previously with much reliability (Cisse et al., 2019; Falasca et al., 2010). Furthermore, our recent analysis has shown that combination indices for S3 and Docetaxel are lower in PC3 cells compared to the other cell lines (data presented in Supplementary Figures 8-10), confirming that this was the most appropriate cell line to use in the in vivo studies.

  1. In Section 2.4., the authors performed colony formation assays to assess the inhibitory activity of the novel S3 inhibitor, only in the murine HiMYC cell line. The study will benefit from similar experiments on the human PCa cell lines. [minor]

Data from clonogenic assays in LNCaP, PC3 and DU145 cells have been added and are presented in Supplementary Figure 7.

  1. Discussion and Conclusion: The authors nicely discuss their findings and put them in context of published data. However, this section needs improvement. Comments:

The Conclusion section should come right after the Discussion and not after the Methods. [minor]

We agree with the Reviewer that conclusions would have been better placed after the Discussion section. This, however, was not our choice as it is the structure provided by Cancers template

  1. The authors comment on the lack of S3 therapeutic efficacy as a single agent their xenograft PC3 model, suggesting that it may be due to a decrease in LPI release rather than lack of Docetaxel intracellular accumulation. The rescue experiment suggested above in Section 2.3. may clarify this issue with experimental data. [minor]

Text has been amended to expand discussion on the potential mechanisms responsible for the increased efficacy of S3 and Docetaxel combination compared to each single treatment.

  1. The authors discuss the implication of Myc in the ABCC1 inhibition response. However, they can only speculate since they do not provide any direct evidence of Myc overexpression in their experimental setup using the human PCa cell lines. As mentioned above, a more detailed characterization of these cell lines in terms of ABCC1 and Myc expression would significantly improve the manuscript.

We agree with the Reviewer that the implication of Myc in the ABCC1 inhibition response is a speculation. We have added a better characterization of cell lines as requested and discussion has been amended accordingly.

Round 2

Reviewer 3 Report

The authors have addressed the majority of reviewer’s comments and the manuscript has significantly improved. However, there is one remaining point that in my opinion require some further clarification:

As requested in comment number 5, the authors provide additional cMyc protein expression data for all the human PCa cell lines used in their study. However, these data indicate that these cell lines do not express similar cMyc levels. In fact, only the PC3 cells exhibit high levels of cMyc, whereas the DU145 and LNCaP have detectable, yet negligible cMyc expression, consistent with previously published data. Of note is that the Tubulin loading control in Supplementary Figure 1 appears to indicate less whole protein present in the DU145 and LNCaP lanes, which may explain the striking difference in cMyc expression between these cell lines. Despite the difference in cMyc expression, the effect of ABCC1 inhibition (by Reversan or S3) in cell survival between the PC3 and LNCaP cells appears to be the same. The authors should further discuss this result, and what it may mean for the implication of cMyc in the inhibition of the ABCC1/LP1 loop.

Author Response

As requested in comment number 5, the authors provide additional cMyc protein expression data for all the human PCa cell lines used in their study. However, these data indicate that these cell lines do not express similar cMyc levels. In fact, only the PC3 cells exhibit high levels of cMyc, whereas the DU145 and LNCaP have detectable, yet negligible cMyc expression, consistent with previously published data. Of note is that the Tubulin loading control in Supplementary Figure 1 appears to indicate less whole protein present in the DU145 and LNCaP lanes, which may explain the striking difference in cMyc expression between these cell lines. Despite the difference in cMyc expression, the effect of ABCC1 inhibition (by Reversan or S3) in cell survival between the PC3 and LNCaP cells appears to be the same. The authors should further discuss this result, and what it may mean for the implication of cMyc in the inhibition of the ABCC1/LP1 loop.

We agree with the Reviewer that our conclusions on c-MYC data are not fully supported by data and decided to recognize this in the revised version of our Discussion. As rightly pointed out by the Reviewer, our Western blotting analysis has revealed different expression levels of c-MYC in PC3, LNCaP and DU145 cells, consistent with data in literature. Nonetheless, these cell lines express similar levels of ABCC1 and they are sensitive to ABCC1 inhibition. A possible explanation for these results is that protein levels of c-MYC, although different between the three cell lines, might still be sufficient to promote ABCC1 expression and activation of the ABCC1/LPI pathway to the same extent in these cells. We recognise, however, that further studies are required to establish for definite whether c-MYC can control ABCC1 expression in prostate cancer cells as it has been shown in other cancer settings. This discussion has been added to the text.

We further discuss our data demonstrating that growth of immortalised epithelial prostate cells induced by overexpression of c-MYC (but not their normal growth) is inhibited by Reversan, clearly suggesting the existence of a link between c-MYC overexpression and the ABCC1/LPI pathway. We suggest that c-MYC overexpression might modulate ABCC1 expression and/or its intracellular localization or it might affect LPI synthesis and/or its release or it might regulate GPR55 expression and/or activity in normal prostate cells and we state clearly that additional work is required to understand the precise link between c-MYC and the ABCC1/LPI/GPR55 autocrine loop.

We have replaced references 20 and 21 and added reference 50. We have corrected minor spell typos. All changes compared to the previous revision are in green.